# Predictors of Human Milk Fatty Acids and Associations with Infant Growth in a Norwegian Birth Cohort

**DOI:** 10.3390/nu14183858

**Published:** 2022-09-17

**Authors:** Rachel L. Criswell, Nina Iszatt, Hans Demmelmair, Talat Bashir Ahmed, Berthold V. Koletzko, Virissa C. Lenters, Merete Å. Eggesbø

**Affiliations:** 1Norwegian Institute of Public Health, Division for Climate and Environmental Health, Nydalen, 0403 Oslo, Norway; 2Skowhegan Family Medicine, Redington-Fairview General Hospital, 46 Fairview Ave, Skowhegan, ME 04976, USA; 3Department of Pediatrics, Dr. von Hauner Children’s Hospital, University Hospital, Ludwig Maximilians Universität Munich, Lindwurmstrasse 4, 80337 Munich, Germany; 4Department of Clinical and Molecular Medicine, Norwegian University of Science and Technology, 7491 Trondheim, Norway

**Keywords:** human milk, obesity, triglyceride fatty acids, infant growth

## Abstract

Triglyceride-bound fatty acids constitute the majority of lipids in human milk and may affect infant growth. We describe the composition of fatty acids in human milk, identify predictors, and investigate associations between fatty acids and infant growth using data from the Norwegian Human Milk Study birth cohort. In a subset of participants (*n* = 789, 30% of cohort), oversampled for overweight and obesity, we analyzed milk concentrations of detectable fatty acids. We modelled percent composition of fatty acids in relation to maternal body mass index, pregnancy weight gain, parity, smoking, delivery mode, gestational age, fish intake, and cod liver oil intake. We assessed the relation between fatty acids and infant growth from 0 to 6 months. Of the factors tested, excess pregnancy weight gain was positively associated with monounsaturated fatty acids and inversely associated with stearic acid. Multiparity was negatively associated with monounsaturated fatty acids and n-3 fatty acids while positively associated with stearic acid. Gestational age was inversely associated with myristic acid. Medium-chain saturated fatty acids were inversely associated with infant growth, and mono-unsaturated fatty acids, particularly oleic acid, were associated with an increased odds of rapid growth. Notably, excessive maternal weight gain was associated with cis-vaccenic acid, which was further associated with a threefold increased risk of rapid infant growth (OR = 2.9, 95% CI 1.2–6.6), suggesting that monounsaturated fatty acids in milk may play a role in the intergenerational transmission of obesity.

## 1. Introduction

Human milk provides nutrients for the optimal development of newborns and infants [1]. The major energy-providing component in human milk is fat, 98–99% of which is triglyceride fatty acids (TGFAs) that are passed into milk from the maternal diet and fat stores or synthesized by the mammary gland [2,3]. The quantity and composition of lipids to which infants are exposed affect growth, vision, inflammatory responses, the gut, and neurodevelopment [4,5,6,7,8,9,10,11,12]. The developmental origins of health and disease hypothesis suggests that these programming effects last beyond infancy, impacting longer-term health outcomes, such as obesity [13].

The importance of human milk TGFAs in determining long-term obesity outcomes is indicated in animal models where exposure to certain TGFAs during early development can change the production and use of lipids within the body [14]. This is supported by human studies that link certain maternal serum fatty acid profiles during pregnancy to infant health outcomes, including blood pressure and neuropsychiatric outcomes, although the evidence for infant growth and obesity is equivocal [8,9,11]. Milk TGFA composition differs across mothers based on geography, sociodemographic factors, and diet and, thus, infants’ exposures vary [15,16,17,18,19,20,21,22,23,24,25,26,27,28,29,30,31,32,33]. Despite the potential impact that exposure to these fatty acids may have on infant health, our knowledge of what determines the composition of milk is still evolving. In this study, we characterized the concentrations and percent composition of detectable TGFAs in human milk and studied the relation between maternal factors and these TGFAs. Further, we explored associations between TGFA milk composition and infant growth during the first six months of life. 

## 2. Materials and Methods

### 2.1. Study Population

This study was based on a subset (*n* = 789, 30% of 2606) from the Norwegian Human Milk Study (HUMIS), a multi-center, Norway-wide cohort of mothers and newborns [34,35,36]. Briefly, new mothers were recruited in 2003–2009 by public health nurses during routine home visits around 2 weeks postpartum in 7 counties across Norway [34]. A subset of mothers was recruited in 2002–2005 by a pediatrician at the maternity ward in Østfold hospital in Southern Norway, two term births for every preterm birth [34,35,36].

Mothers were asked to save and freeze a hand- or pump-expressed sample of the morning’s first milk prior to infant feeding on each of 8 consecutive days between 2 weeks and 2 months postpartum [35]. Milk samples were sent to the Norwegian Institute of Public Health in Oslo, Norway, where they were frozen and stored. The milk was collected in accordance with protocols established by the World Health Organization for biomonitoring of chemicals in human milk [37]. Mothers completed questionnaires about the child’s weight and height, maternal weight, and relevant confounders. Data from the Norwegian Medical Birth Registry and mothers’ medical pregnancy journals were used to supplement information on participants. 

The study subset was determined as illustrated in Figure 1. Singleton mothers who were overweight and obese (pre-pregnancy body mass index (BMI) > 30 kg/m^2^ or BMI at delivery >32 kg/m^2^ with a pre-pregnancy BMI of <27 kg/m^2^) were oversampled and represent 41.5% of the sample as compared to 34.0% of women in the whole HUMIS cohort [38].

### 2.2. TGFA Composition Analysis

TGFAs were measured in pooled (per mother) milk samples collected at median 31 days (range: 14–108 days). Milk samples were shipped by air freight on dry ice to LMU Munich for analysis. TGFA composition was measured by gas chromatography (GC) as previously described [39]. Briefly, 100 µL milk was combined with 700 µL methanol and 600 µL tert-butyl-methyl-ether, vortexed for 30 seconds, and centrifuged at room temperature with 1500 g for 5 minutes to precipitate proteins. Next, 300 µL supernatant with the dissolved lipids were transferred into a 4 mL glass vial and combined with 1260 µL methanol, 140 µL hexane, and 400 µL methanolic hydrochloric acid. The content was mixed by shaking and heated to 100 °C for one hour to synthesize fatty acid methyl esters from free and lipid-bound fatty acids. Water (500 µL) and hexane (1000 µL) were added to achieve phase separation. An aliquot of the organic upper phase was used for GC after appropriate dilution for injection. GC separation of fatty acid methyl esters on a 60 m BPX-70 column and detection by flame ionization were performed as described elsewhere, with the modification that tri-undecanoin instead of tri-pentadecanoin was used as the internal standard for quantification [40]. TGFAs are presented as absolute concentrations (g/L) and percent of total fatty acid concentration by weight (%).

Thirty-two TGFAs with chain lengths of 8–22 carbon atoms could be quantified. As a control for analytical quality, 82 aliquots of a reference milk sample were distributed randomly among the study samples for the analysis. For TGFA concentrations, coefficients of variation were between 6% (C12:0) and 33% (C22:5n-6), with an average of 13%. For the percentage contributions of TGFAs, coefficients of variation ranged from 1% (C16:0) to 33% (C22:5n-6), with an average of 9%.

### 2.3. Predictors of TGFA Composition

We studied percent composition of 32 TGFAs, total saturated fatty acids (SFA), polyunsaturated fatty acids (PUFA), n-3 PUFAs, n-6 PUFAs, monounsaturated fatty acids (MUFA), trans-fatty acids, medium chain fatty acids (MCFA), and long-chain PUFAs (LC-PUFA). In addition, we studied SFA/PUFA and n-6/n-3 PUFA ratios (Table 1). For the purposes of this analysis, the intermediate-chain-length FA myristic acid (C14:0) was categorized as an MCFA due to its synthesis in the mammary gland and similar effects as MCFAs in the infant diet [1,41,42].

We identified potential predictors of TGFAs from the literature [22,30,31,43,44,45,46,47,48] and included pre-pregnancy BMI (underweight versus normal/overweight versus obese) [49], excess weight gain in pregnancy (categorical) [50], parity (primiparous versus multiparous), smoking at the start of pregnancy (never/former smoker versus current smoker), gestational age at birth (days), delivery mode (vaginal versus cesarean section), fatty fish dinners in one year (continuous, in servings), and cod liver oil consumption in one year (continuous, in servings). 

We identified covariates from the literature [15,16,17,18,19,20,21,22,23,24,25,26] and identified the minimally sufficient adjusted set for each predictor model using directed acyclic graphs (DAGs) [51]. Dependent on each predictor’s DAG, confounders included a combination of pre-pregnancy BMI, excess weight gain in pregnancy, parity, gestational age, smoking, maternal education (<12, 12, or >12 years), maternal age (years), and birthweight (grams) (Appendix A). We adjusted all models for infant age at time of milk collection (days) and whether formula had been introduced at time of milk collection (yes/no). 

### 2.4. TGFA Associations with Infant Growth

The primary outcome for these analyses was infant growth defined as change in weight-for-age *z*-score from 0–6 months. Weight and height data were recorded in children’s health cards during regular health examinations and were parent reported in questionnaires at 1, 6, 12, and 24 months. Using these weight and height data, we estimated weight at exactly 6 months in a sex-specific multilevel (mixed) linear model fitted with cubic polynomials and random effects for infant [52]. For the period of 0–6 months, infants had up to 5 weight and height data points. For this time period, the distribution of rapid growth was no different between infants with less than 2 weight points available and those in the whole cohort. We calculated sex- and gestational-age-specific weight-for-age *z*-scores for birth and 6 months for the full HUMIS population (*n* = 2606) [53]. We then calculated the primary outcome change in weight-for-age z-score as the difference between birth and 6-month *z*-scores (continuous). As a secondary outcome, we evaluated rapid growth, a risk factor for obesity later in life [54,55], defined as a change in *z*-score greater than 0.67 (normal versus rapid) [52].

Exposures were TGFA percent composition and ratios. Again, we identified appropriate potential confounders through the literature and chose the minimally sufficient adjusted set identified in DAGs (Appendix A). We adjusted all growth models for maternal age, smoking, education, pre-pregnancy BMI (continuous), gestational weight gain (continuous), parity, and sex. 

### 2.5. Statistical Analysis

We summarized the data (exposure, covariate, and outcome variables) in the complete case data set. We imputed missing data among predictors (2–4%), covariates (13–16%), and growth outcomes (1%) using multiple imputation by chained equations [56]. There were no missing data among the TGFA variables. We used the STATA 14.0 “mi impute chained” function with a burn-in of 1000 to generate 10 imputed data sets. 

For the predictor models, we used linear regressions for each predictor with TGFA percent composition and ratios as outcomes. We investigated the relation between percent composition of TGFAs and change in weight-for-age z-score using linear regression and with rapid growth using logistic regression. We ran all models using imputed data sets and the “mi estimate” STATA 14.0 function and Rubin’s rules to combine imputed data set results [56]. We present results as β-coefficients or odds ratios (ORs) with 95% confidence intervals (CI). All regression models were adjusted for the confounders described above.

### 2.6. Sensitivity Analyses

For TGFAs that were inversely associated with rapid growth, we ran a sensitivity analysis in the complete case set, testing for association with failure to thrive, defined as change in weight-for-age *z*-score < −0.67 [57]. Additionally, we assessed the sensitivity of our results by testing the complete case data.

We assessed all regression models for heteroscedasticity and normality of residuals and used the Hosmer–Lemeshow’s test to establish goodness of fit for the logistic regression models. We assessed the combination of high leverage and residuals to fit regression models with and without influential observations. 

## 3. Results 

### 3.1. Study Population and Milk Composition

Of 789 mothers, 41.5% were overweight or obese and 57.0% had excess pregnancy weight gain, while 18.2% of infants were rapid growers (Table 2). Comparison of our study subsample to the larger cohort and the general Norwegian population can be found in Appendix A. The most abundant TGFAs were palmitic acid (C16:0), oleic acid (C18:1n-9), and linoleic acid (C18:2n-6) (Table 3, Figure 2). 

### 3.2. Associations between Predictors and Fatty Acid Composition

We found that the following maternal factors were associated with fatty acid composition: gestational age, cod liver oil intake, fatty fish intake, excess weight gain in pregnancy, parity, and obesity (Table 4). We found no association between maternal smoking, underweight BMI, or mode of delivery and TGFA percent composition (Appendix A). 

### 3.3. Associations between TGFA Composition and Infant Growth

MCFAs, lauric acid (C12:0), and myristic acid (C14:0) were inversely associated with infant growth and rapid growth. MUFAs, palmitoleate, oleic acid, and cis-vaccenic acid were positively associated with infant growth and an increased odds of rapid growth (Table 5, Figure 3 and Figure 4). SFAs and SFA/PUFA were inversely associated with infant growth and odds of rapid growth (Table 4, Appendix A). None of the TGFAs analyzed were associated with failure to thrive (Appendix A). 

The non-heteroscedasticity and normality of residual assumptions of our models held. Effect estimates were comparable in models with and without the inclusion of extreme values and outliers, from analyses with the complete case set and multiply imputed sets and in unadjusted and adjusted models. Due to the large number of analyses, some of our results may have been random findings.

## 4. Discussion

We found that gestational age, maternal obesity, excess pregnancy weight gain, parity, and fatty fish and cod liver oil intake were associated with TGFA composition in human milk. Medium-chain saturated fatty acids were inversely associated with infant growth. MUFAs, including oleic acid, palmitoleate, and cis-vaccenic acid, were all associated with increased odds of rapid growth.

### 4.1. Medium-Chain Fatty Acids

The mammary gland secretes MCFAs directly into milk, rather than mobilizing them from maternal diet or adipose tissue. Milk TGFAs are typically SFAs, with carbon chains ranging from 8 to 14 atoms. Among adults, MCFAs are associated with increased cholesterol as a result of their roles in desaturase [58] and lipoprotein regulation [59]. However, among infants, they are an important source of energy because of their relatively shorter chain length as compared to LC-PUFAs and unique absorption pattern through the liver, independent of mitochondrial transport pathways [60,61]. MCFAs also play a role in developing the infant gut microbiome and have antibacterial and antiviral properties [60]. 

MCFAs are proportionally higher in the milk of mothers of premature and low-birth-weight infants and they are easier for these infants with immature gastrointestinal tracts to utilize than longer-chain TGFAs [18,21,62]. Our findings suggest a dose–response relation between gestational age and myristic acid that continues through the whole gestational spectrum, including among term babies. 

We found an inverse, though non-significant, relation between MCFAs in human milk and infant growth. As noted above, compared to LC-PUFAs, MCFAs are more readily oxidized for energy rather than being stored in adipose tissue for later use, resulting in less weight gain [61,63]. In animal models, breastfeeding pups of mice fed high-MCFA diets of virgin coconut oil showed significantly lower body weight [64]. 

### 4.2. Monounsaturated Fatty Acids

Excess maternal weight gain in pregnancy was associated with increased palmitoleate and cis-vaccenic acid and inversely associated with stearic acid, expanding on the findings of previous studies [16]. The stearoyl-CoA desaturase-1 (SCD1) pathway converts palmitic acid to palmitoleate, which is then elongated to cis-vaccenic acid. Stearic acid competes with the SCD1 pathway for palmitic acid as it is elongated from palmitic acid [65]. SCD1 activity is a marker for obesity in adults [65,66,67] and might also be increased in women with excess weight gain in pregnancy. If this was the case, increased SCD1 activity might explain increased palmitoleate and cis-vaccenic acid and concurrent decreased stearic acid associated with excessive pregnancy weight gain.

A recent study in the CHILD cohort in Canada had opposite findings, with maternal obesity associated with decreased overall combined MUFAs [30]. This difference in results might reflect different dietary fat choices among obese women in Norway and Canada, respectively, as the CHILD cohort had a higher percent composition of oleic acid as compared to our cohort.

Multiparity showed inverse associations for these MUFAs, even when controlling for maternal BMI. The majority of MUFAs and PUFAs in milk derive from maternal adipose tissue [68], and women with previous pregnancies and lactation periods may have depleted stores with each successive pregnancy. Since stearic acid is highly represented in dietary sources, it may not deplete so readily and, therefore, be more abundant in these women’s milk [21]. 

Infants exposed to higher proportions of MUFAs, palmitoleate, oleic acid, and cis-vaccenic acid had increased odds of rapid growth. Given the high percent composition of oleic acid in our cohort as compared to the other MUFAs, variation within the IQR of this TGFA is associated with an almost fourfold increase in odds of rapid growth among infants. Our results replicate in a larger sample size those of an Australian study that found associations between oleic acid and cis-vaccenic acid and infant growth at 6 months among 18 mother–infant dyads [12]. Among adults, increased levels of circulating oleic acid have been found to be associated with metabolic syndrome and cardiovascular disease [10,67]. Further, in rats, a high MUFA/SFA ratio in maternal milk results in increased offspring adiposity, particularly when those MUFAs include oleic acid, palmitoleate, and cis-vaccenic acid [69].

MUFAs may play a role in mediating (rather than confounding) the relation between maternal obesity and infant obesity, although human milk protects against obesity compared with other infant feeding methods [70,71]. While controversial, some studies suggest that circulating palmitoleate may actively affect metabolism, acting as a “lipokine”, exerting a hormone-like influence [72,73]. Other studies have linked elevated levels of circulating palmitoleate to increased cell proliferation in adipose tissue in small-for-gestational-age mice [69]. Alternatively, high levels of oleic acid, palmitoleate, and cis-vaccenic acid may be markers for other pro-growth components highly represented in the milk of obese women [65,67]. Taken together, our results suggest that maternal obesity and metabolism may play a role in infant MUFA exposure and, therefore, growth. Clinical interventions targeting appropriate gestational weight gain may help to reduce risk factors for childhood obesity.

### 4.3. Polyunsaturated Fatty Acids

N-3 fatty acids have robust positive health effects, and the main dietary source is cold-water fish [74]. We found a positive association between fatty fish and cod liver oil intake and n-3 fatty acids levels in milk, replicating previous studies [2,3,19,24,30,31,75,76]. However, from a public-health standpoint, the positive role of dietary cod liver oil and fatty fish on n-3 fatty acid levels in human milk must be balanced with the risk of increased fat-soluble toxicant levels in human milk that may accompany high fish intake. Exposure to fat-soluble toxicants in human milk has been associated with obesity in children [52,53,77,78,79,80].

We found that maternal obesity had inverse associations with EPA, DHA, and total n-3 PUFAs. In Sweden, obese mothers also had lower levels of milk n-3 PUFAs than normal-weight mothers [81], and results from the French EDEN cohort suggest that obesity is associated with decreased DHA and ALA in colostrum [31]. Dietary intake is the major determinant of milk levels of EPA and DHA [27,82], while the majority of other PUFAs in milk derive from adipose tissue [68,83]. It is possible that in the milk of obese women, fats from maternal stores displace dietary fats, which may already be low due to diets with fewer n-3 PUFAs.

Parity was inversely associated with EPA and DHA. Dutch studies have documented maternal depletion of DHA over the course of pregnancy, and multiparity has been associated with decreased maternal serum DHA [84,85], although not at one year after pregnancy [48]. The EDEN cohort also found increased DHA in the colostrum of primiparous as compared to multiparous women [31]. Conversely, the CHILD cohort did not find any association between parity and TFGAs in milk [30]. Although there may be some depletion of EPA and DHA from maternal fat stores over the course of pregnancy and lactation, the relation between parity and EPA and DHA remains unclear. Future research could investigate the relation between inter-pregnancy interval and these TGFAs in milk and optimal n-3 PUFAs levels for multiparous mothers.

We found no association between n-3 PUFAs and infant growth, which supports the findings from a study of Australian women that found no significant association between EPA and DHA and infant growth [12]. While some studies suggest associations between these TGFAs and metabolic outcomes in adults and pregnant people [10,65], previous research among infants has suggested that these TGFAs may play a more substantial role in infant neurodevelopment [4,7,9].

### 4.4. Strengths and Limitations

To our knowledge, ours is among the first studies to assess such a broad array of TGFAs in association with maternal predictors and growth outcomes. Our study is strengthened by a large sample size. Due to linkage with the Norwegian Medical Birth Registry, variables that are often subject to recall bias may have been reported with more precision than in other studies. Finally, our study is generalizable to the Norwegian population, as our sample population had similar characteristics to the general population of women who had recently given birth (Appendix A). While Norway has lower obesity rates than many other nations [86], oversampling for obesity increased the power of our analysis.

Our study had some limitations. First, the timeframe for milk sampling was wide, ranging from 14 to 108 days. However, all but two subjects collected milk samples between 14 and 60 days. Furthermore, all samples represented mature milk, so this range should not negatively affect our accuracy in reporting concentrations and percent composition. The concentrations and patterns of composition in our study are consistent with European and global averages [27,30,44,87,88]. The inclusion of women and infants who had combined human milk and formula feeding at the time of milk collection could have affected our results, in that different infant feeding patterns may change milk composition, and the nutritional content of formula may affect infant growth. We addressed this in our predictor model by adjusting for formula introduction. Associations between TGFAs and growth with and without adjustment for formula were not materially different, so inclusion of combination feeders should not reduce the validity of our results. Furthermore, a huge majority of the mothers was exclusively breastfeeding at the time of milk sampling.

We included fatty fish and cod liver oil intake as potential predictors of n-3 fatty acid composition, but lacked more comprehensive dietary exposures, which can be important contributors for other TGFAs in milk. There is evidence of a relation between fatty acid desaturase (FADS) gene expression and milk TGFA composition [30], and we did not have FADS profiles. Additionally, we did not have data on milk volume, total energy, or other energy-generating nutrients, such as protein concentrations in our milk samples, which can be factors that affect infant growth. As a result, unmeasured dietary or genetic variables may confound some of our findings.

Finally, we found very wide confidence intervals when running logistic regressions of TGFAs’ association with rapid growth, suggesting that logistic regression may not optimally model the association between the exposure and outcome. However, our findings do suggest patterns in TGFA exposure and growth outcomes that are corroborated by the linear regression findings.

## 5. Conclusions

In conclusion, our study found that excess weight gain in pregnancy and parity are associated with increased MUFAs and decreased n-3 PUFA content in milk. Importantly, exposure to MUFAs in breast milk was associated with obesity outcomes in infants. Our results indicate that targeting excessive pregnancy weight gain could be an effective intervention to prevent childhood obesity.

## Figures and Tables

**Figure 1 nutrients-14-03858-f001:**
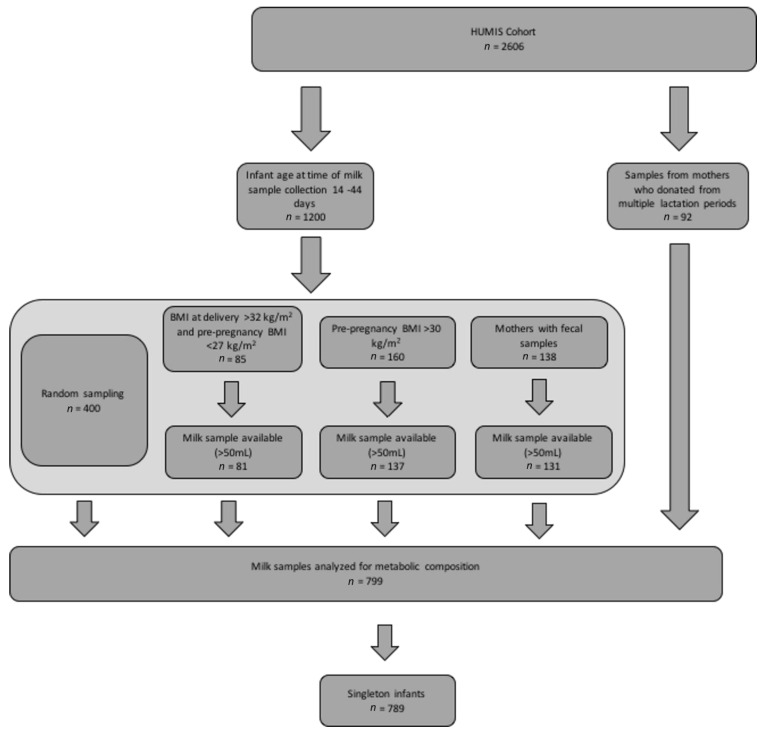
Participant flow diagram for present analysis. Legend: This figure describes the sampling process by which the milk samples for the present analysis (*n* = 789) were selected from the Norwegian Human Milk Study (HUMIS) cohort (*n* = 2606). BMI stands for body mass index.

**Figure 2 nutrients-14-03858-f002:**
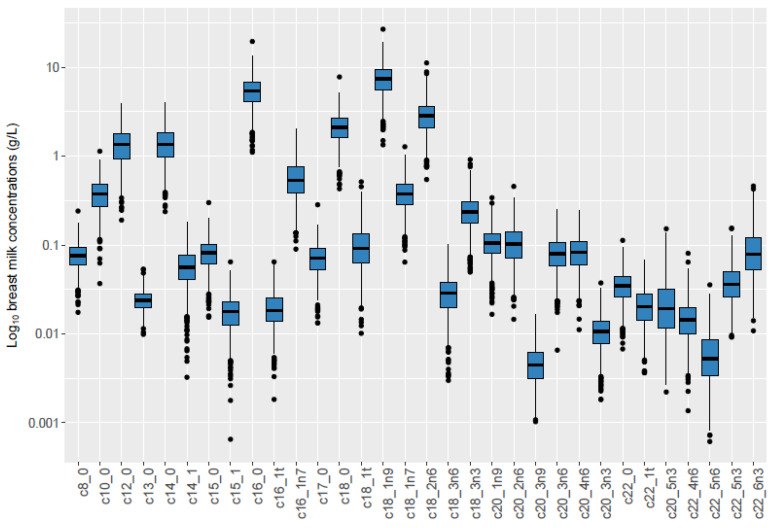
Boxplot of TGFA concentrations in study population (*n* = 789) ^1^. Legend: ^1^ X-axis represents TGFAs measured in milk samples. Y-axis represents log_10_ of breast milk concentration.

**Figure 3 nutrients-14-03858-f003:**
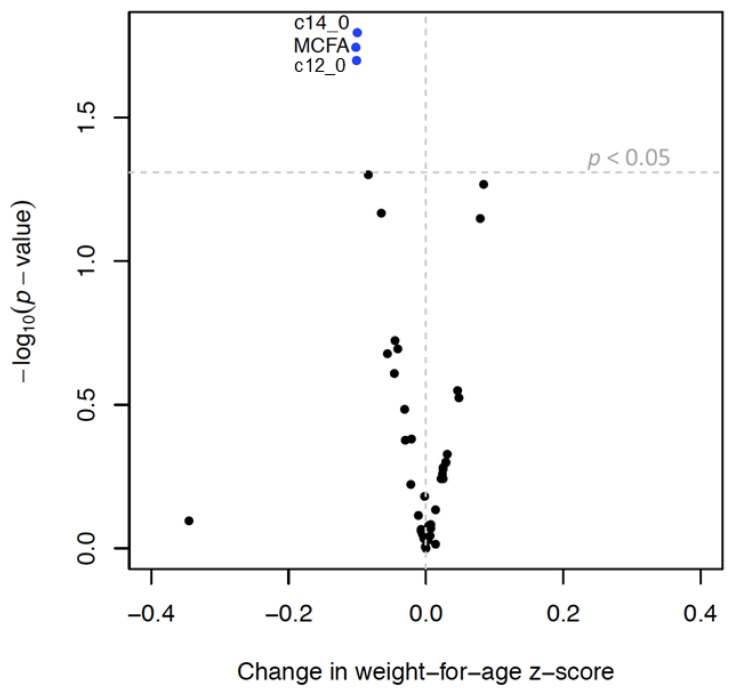
Volcano plot ^1^ showing associations ^2^ between triglyceride fatty acid (TGFA) composition in milk and infant growth ^3^. Legend: **^1^** Each dot represents change in weight-for-age z-score (x axis) per interquartile range (IQR) increase in TGFA composition. Y-axis represents the −log_10_ of the *p*-value. The vertical dotted line represents a beta coefficient of 0 and the horizontal dotted line represents a *p*-value of 0.05, so that dots above that line represent associations that have a *p*-value < 0.05. ^2^ Models were run in the multiply imputed data set (*n* = 789) in 10 imputed sets and adjusted for maternal age, smoking, education, pre-pregnancy body mass index, gestational weight gain, parity, and child sex. ^3^ Infant growth defined as change in weight-for-age z-score between 0 and 6 months.

**Figure 4 nutrients-14-03858-f004:**
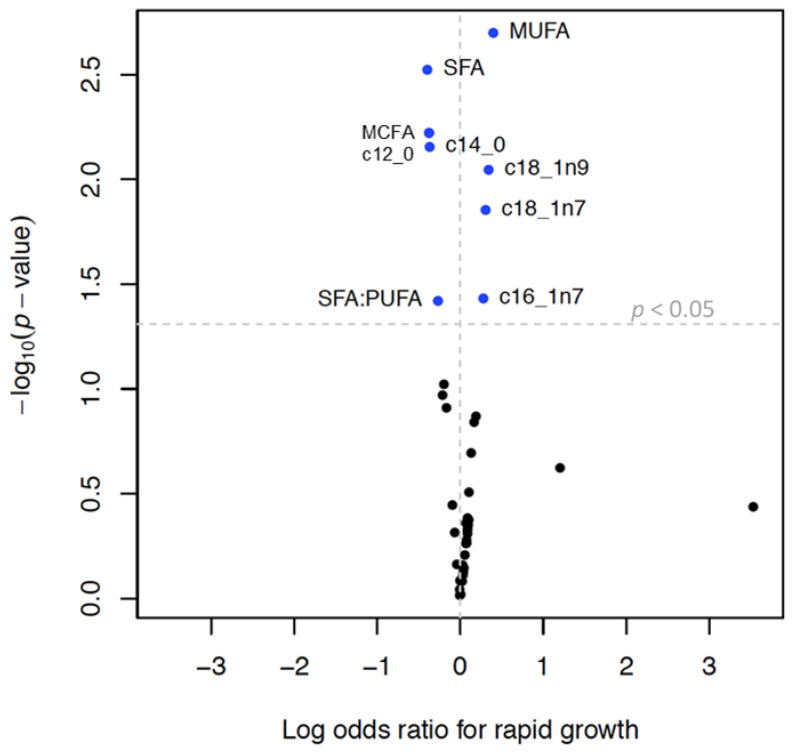
Volcano plot ^1^ showing associations ^2^ between triglyceride fatty acid (TGFA) composition in milk and rapid growth ^3^ Legend: **^1^** Each dot represents change in the log of the odds ratio (OR) for rapid growth (x axis) per interquartile range (IQR) increase in TGFA. Y-axis represents the −log_10_ of the *p*-value. The vertical dotted line represents an OR of 1 and the horizontal dotted line represents a *p*-value of 0.05, so that dots above that line represent associations that have a *p*-value < 0.05. ^2^ Models were run in the multiply imputed data set (*n* = 789) in 10 imputed sets and adjusted for maternal age, smoking, education, pre-pregnancy body mass index, gestational weight gain, parity, and child sex. ^3^ Defined as change in weight-for-age z-score between 0 and 6 months > 0.67.

**Table 1 nutrients-14-03858-t001:** TGFAs included in each fatty acid sub-category ^1^.

Category	TGFAs in Each Category
Saturated fatty acids (SFA) ^2^	C8:0, C10:0, C12:0, C13:0, C14:0, C15:0, C16:0, C17:0, C18:0, C22:0
Medium chain fatty acids (MCFA)	C8:0, C10:0, C12:0, C14:0 ^3^
Polyunsaturated fatty acids (PUFA)	C18:3n-3, C20:3n-3, C20:5n-3, C22:5n-3, C22:6n-3 C18:2n-6, C18:3n-6, C20:2n-6 ^4^, C20:3n-6, C20:4n-6, C22:4n-6, C22:5n-6 C20:3n-9
N-3 PUFA	C18:3n-3, C20:3n-3, C20:5n-3 C22:5n-3, C22:6n-3
N-6 PUFA	C18:2n-6, C18:3n-6 C20:2n-6, C20:3n-6, C20:4n-6 C22:4n-6, C22:5n-6
Long chain PUFA (LC-PUFA)	C20:3n-3, C20:3n-6, C20:3n-9, C20:4n-6, C20:5n-5 C22:4n-6, C22:5n-6, C22:5n-3, C22:6n-3
Monounsaturated fatty acids (MUFA)	C14:1, C15:1, C16:1n-7, C18:1n-9, C18:1n-7, C20:1n-9
Trans fatty acids	C16:1t, C18:1t, C22:1t

^1^ Please note that individual triglyceride fatty acids (TGFAs) can appear in more than one category. ^2^ As the concentration of C11:0 was used as an internal standard, its concentration was not measured. ^3^ As the classification of myristic acid (C14:0) is ambiguous, for the purposes of this analysis, myristic acid is characterized as a medium-chain fatty acid (MCFA) due to its production in the mammary gland along with caprylic acid (C8:0), undecanoic acid (C10:0), and lauric acid (C12:0) and similar effects as other MCFAs in the diets of infants. ^4^ As this TGFA is an elongation of linoleic acid (rather than a downstream product of a desaturase enzyme), it has been excluded from the long-chain polyunsaturated fatty acid (LC-PUFA) classification.

**Table 2 nutrients-14-03858-t002:** Characteristics of study population (*n* = 789).

Maternal or Infant Characteristic	N (%) or Median (IQR)	Missing
Pre-pregnancy body mass index (kg/m^2^)	24.0 (21.6–27.5)	28
Underweight (<18.5)	34 (4.5)	
Normal weight (18.5–24.9)	411 (54.0)	
Overweight (25–29.9)	210 (27.6)	
Obese (>30)	106 (13.9)	
Gestational weight gain (kg)	15.0 (11.0–18.5)	33
Excess weight gain in pregnancy ^1^		38
No	323 (43.0)	
Yes	428 (57.0)	
Maternal age at pregnancy start (years)	30.0 (26.0–33.0)	0
Parity		20
Primiparous	306 (39.8)	
Multiparous	463 (60.2)	
Smoking ^2^		162
Never/former smoker	554 (88.4)	
Current smoker	73 (11.6)	
Education		8
<12 years	71 (9.1)	
12 years	105 (13.4)	
>12 years	605 (77.5)	
Total fatty fish dinners in one year (servings)	17.1 (7.1–36.0)	58
Total cod liver oil intake in one year (servings)	52 (0–364)	45
Gestational age (days)	282 (275–288)	3
Infant birthweight (g)	3670 (3280–4030)	0
Delivery mode		0
Vaginal delivery	667 (84.5)	
Cesarean section	122 (15.5)	
Infant age at milk collection (days)	31 (25–36)	101
Formula introduced at time of milk collection		128
No	603 (91.2)	
Yes	58 (8.8)	

^1^ Defined per body mass index category according to Institutes of Medicine guidelines: for underweight women, >18.1 kg; for normal weight women, >15.9 kg; for overweight women, >11.3 kg; and for obese women, >9.1 kg. ^2^ Defined as smoking at the start of pregnancy.

**Table 3 nutrients-14-03858-t003:** Concentrations and percentages of fatty acids in milk (*n* = 789).

Fatty Acid		Absolute Concentration (g/L)	Percent of Total Fatty Acid Content (%)
		Median (IQR)	Median (IQR)
**Saturated fatty acids**		10.91 (8.40–13.92)	47.0 (44.6–49.5)
C8:0 ^1^	Caprylic acid	0.08 (0.06–0.09)	0.3 (0.3–0.4)
C10:0 ^1^	Capric acid	0.37 (0.27–0.48)	1.6 (1.4–1.9)
C12:0 ^1^	Lauric acid	1.36 (0.93–1.78)	5.8 (4.7–6.9)
C13:0	Tridecanoic acid	0.02 (0.02–0.03)	0.1 (0.1–0.1)
C14:0 ^1^	Myristic acid	1.36 (0.98–1.81)	5.9 (5.0–6.9)
C15:0	Pentadecanoic acid	0.08 (0.06–0.10)	0.4 (0.3–0.4)
C16:0	Palmitic acid	5.37 (4.11–6.82)	23.1 (22.0–24.2)
C17:0	Margaric acid	0.07 (0.05–0.09)	0.3 (0.3–0.3)
C18:0	Stearic acid	2.10 (1.61–2.68)	9.1 (8.4–9.9)
C22:0	Behenic acid	0.04 (0.03–0.04)	0.2 (0.1–0.2)
**Monounsaturated fatty acids**		8.60 (6.39–10.81)	36.7 (34.8–38.7)
C14:1	Myristoleic acid	0.06 (0.04–0.08)	0.3 (0.2–0.3)
C15:1	Pentadecanoic acid	0.02 (0.01–0.02)	0.1 (0.1–0.1)
C16:1n-7	Palmitoleate	0.53 (0.38–0.75)	2.4 (2.0–2.9)
C18:1n-9	Oleic acid	7.42 (5.57–9.35)	31.8 (30.1–33.6)
C18:1n-7	Cis-vaccenic acid	0.38 (0.28–0.48)	1.6 (1.5–1.8)
C20:1n-9	Eicosenoic acid	0.11 (0.08–0.14)	0.4 (0.4–0.5)
**Trans fatty acids**		0.13 (0.10–0.19)	0.6 (0.5–0.7)
C16:1t	Palmitoleic acid	0.02 (0.01–0.03)	0.1 (0.1–0.1)
C18:1t	Elaidic acid	0.09 (0.06–0.13)	0.4 (0.3–0.5)
C22:1t	Erucic acid	0.02 (0.01–0.03)	0.1 (0.1–0.1)
**Polyunsaturated fatty acids**		3.59 (2.66–4.53)	15.4 (13.6–17.4)
C20:3n-9 ^2^	Mead acid	<0.01 (<0.01–0.01)	<0.1 (<0.1–<0.1)
**N-3 polyunsaturated fatty acids**		0.39 (0.29–0.52)	1.7 (1.4–2.1)
C18:3n-3	Alpha-linoleic acid	0.24 (0.18–0.31)	1.0 (0.9–1.2)
C20:3n-3 ^2^	Eicosatrienoic acid	0.01 (0.01–0.01)	0.1 (<0.1–0.1)
C20:5n-3 ^2^	Eicosapentaenoic acid	0.02 (0.01–0.03)	0.1 (0.1–0.1)
C22:5n-3 ^2^	Docosapentaenoic acid	0.04 (0.03–0.05)	0.2 (0.1–0.2)
C22:6n-3 ^2^	Docosahexaenoic acid	0.08 (0.05–0.12)	0.3 (0.2–0.5)
**N-6 polyunsaturated fatty acids**		3.18 (2.34–4.00)	13.7 (11.8–15.4)
C18:2n-6	Linoleic acid	2.83 (2.09–3.61)	12.2 (10.5–13.9)
C18:3n-6	Gamma-linoleic acid	0.03 (0.02–0.04)	0.1 (0.1–0.2)
C20:2n-6	Eicosadienoic acid	0.10 (0.07–0.14)	0.4 (0.4–0.6)
C20:3n-6 ^2^	Dihomo-gamma-linoleic acid	0.08 (0.06–0.11)	0.3 (0.3–0.4)
C20:4n-6 ^2^	Arachidonic acid	0.08 (0.06–0.11)	0.4 (0.3–0.4)
C22:4n-6 ^2^	Adrenic acid	0.01 (0.01–0.02)	0.1 (0.1–0.1)
C22:5n-6 ^2^	Osbond acid	0.01 (<0.01–0.01)	<0.1 (<0.1–<0.1)
Long chain polyunsaturated fatty acids ^3^		0.35 (0.26–0.46)	1.5 (1.3–1.7)
Medium chain fatty acids ^4^		3.16 (2.28–4.14)	13.6 (11.6–16.0)
Saturated fatty acids/Polyunsaturated fatty acids		3.04 (2.59–3.60)	--
N-6/n-3 polyunsaturated fatty acids		8.17 (6.81–9.46)	--
Total fatty acids		23.44 (18.03–29.40)	--

^1^ Here also classified as a medium-chain fatty acid. ^2^ Here also classified as a long-chain-polyunsaturated fatty acid. ^3^ This category includes C20:3n-3, C20:3n-6, C20:3n-9, C20:4n-6, C20:5n-3, C22:4n-6, C22:5n-6, C22:5n-3, and C22:6n-3. ^4^ This category includes C8:0, C10:0, C12:0, and C14:0. As the classification of myristic acid (C14:0) is ambiguous, for the purposes of this analysis, myristic acid is characterized as an MCFA due to its production in the mammary gland along with caprylic acid (C8:0), undecanoic acid (C10:0), and lauric acid (C12:0) and similar effects as other MCFAs in the diets of infants.

**Table 4 nutrients-14-03858-t004:** Notable maternal predictors of triglyceride fatty acid percent composition in milk from Norwegian mothers ^1^.

Predictor	Fatty Acid		β-Coefficient (95% Confidence Interval)
Gestational age ^2^	C14:0	Myristic acid	−0.02 (−0.02–−0.01)
	Monounsaturated fatty acids		0.03 (0.02–0.05)
Cod liver oil intake ^3^	C15:0	Pentadecanoic acid	0.0001 (^4^)
	C20:1n-9	Eicosenoic acid	0.0001 (^4^)
	C20:5n-3	Eicosapentaenoic acid	0.0002 (^4^)
	C22:1t	Erucic acid	0.00004 (^4^)
	C22:4n-6	Adrenic acid	−0.00002 (^4^)
	C22:5n-3	Docosapentaenoic acid	0.0001 (^4^)
	C22:6n-3	Docosahexaenoic acid	0.001 (<0.001–0.001)
	N-3 polyunsaturated fatty acids		0.001 (^4^)
	Long chain polyunsaturated fatty acids		0.001 (<0.001–0.001)
	n-6/n-3 polyunsaturated fatty acids		−0.004 (−0.01–−0.003)
Fatty fish intake ^5^	C20:1n-9	Eicosenoic acid	0.001 (^4^)
	C20:5n-3	Eicosapentaenoic acid	0.001 (<0.001–0.001)
	C22:1t	Erucic acid	0.0002 (^4^)
	C22:5n-3	Docosapentaenoic acid	0.001 (<0.001–0.001)
	C22:6n-3	Docosahexaenoic acid	0.002 (0.001–0.002)
	N-3 polyunsaturated fatty acids		0.004 (0.003–0.01)
	Long chain polyunsaturated fatty acids		0.003 (0.002–0.004)
	n-6/n-3 polyunsaturated fatty acids		−0.02 (−0.02–−0.01)
Excess pregnancy weight gain ^6^	C16:1n-7	Palmitoleate	0.27 (0.19–0.36)
	C18:0	Stearic acid	−0.42 (−0.61–−0.23)
	C18:1n-7	Cis-vaccenic acid	0.10 (0.07–0.13)
	Medium chain saturated fatty acids		−1.10 (−1.64–−0.55)
Parity ^7^	C16:1n-7	Palmitoleate	−0.19 (−0.28–−0.09)
	C18:0	Stearic acid	0.50 (0.30–0.71)
	C18:1n-7	Cis-vaccenic acid	−0.07 (−0.11–−0.04)
	C20:5n-3	Eicosapentaenoic acid	−0.02 (−0.04–−0.01)
	C22:6n-3	Docosahexaenoic acid	−0.08 (−0.12–−0.05)
Obesity ^8^	C20:5n-3	Eicosapentaenoic acid	−0.03 (−0.05–−0.02)
	C22:6n-3	Docosahexaenoic acid	−0.10 (−0.15–−0.06)
	N-3 polyunsaturated fatty acids		−0.22 (−0.32–−0.11)

^1^ Models run in 10 multiply imputed data sets (*n* = 789). This table includes results with relatively larger effect estimates or small confidence intervals. Due to the large number of associations examined, some findings may be due to chance. ^2^ Defined in days. Model adjusted for body mass index, smoking, parity, infant age at milk collection, formula introduction. ^3^ Defined as total servings cod liver oil in previous year. Model adjusted for education, body mass index, fatty fish intake, infant age at milk collection, formula introduction. ^4^ Represents a CI with lower and upper limits <0.001. ^5^ Defined as total number fatty fish dinners in previous year. Model adjusted for body mass index, education, cod liver oil, infant age at milk collection, formula introduction. ^6^ Defined per body mass index category according to Institutes of Medicine guidelines: for underweight women, >18.1 kg; for normal weight women, >15.9 kg; for overweight women, >11.3 kg; and for obese women, >9.1 kg. Model adjusted for gestational age, maternal age, education, parity, infant age at milk collection, formula introduction. ^7^ Defined as primiparous vs. multiparous. Model adjusted for maternal age, education, infant age at milk collection, formula introduction. ^8^ Defined as body mass index >30 kg/m^2^ as per World Health Organization guidelines. Model adjusted for maternal age, education, parity, infant age at milk collection, formula introduction. β-coefficient shows effect estimate as compared to normal/overweight women.

**Table 5 nutrients-14-03858-t005:** Associations between triglyceride fatty acid percent composition ^1^ in human milk and infant growth.

Fatty Acid		Infant Growth ^2^		Rapid Growth ^3^	
		β-Coefficient (95% CI)	*p*-Value	Odds Ratio (95% CI)	*p*-Value
**Saturated fatty acids**		−0.02 (−0.04–<0.01) *	0.05	0.92 (0.87–0.97) *	0.003 *
C8:0 ^4^	Caprylic acid	−0.27 (−0.93–0.39)	0.42	0.17 (0.02–1.36)	0.10
C10:0 ^4^	Capric acid	−0.11 (−0.29–0.06)	0.21	0.65 (0.38–1.10)	0.12
C12:0 ^4^	Lauric acid	−0.05 (−0.08–−0.01) *	0.02 *	0.85 (0.75–0.95) *	0.006 *
C13:0	Tridecanoic acid	−0.62 (−1.86–0.62)	0.33	0.04 (<0.01–2.45)	0.12
C14:0 ^4^	Myristic acid	−0.06 (−0.10–−0.01) *	0.02 *	0.82 (0.71–0.95) *	0.007 *
C15:0	Pentadecanoic acid	−0.06 (−0.76–0.64)	0.86	1.46 (0.19–11.2)	0.72
C16:0	Palmitic acid	<0.01 (−0.04–0.04)	0.97	0.94 (0.84–11.2)	0.24
C17:0	Margaric acid	−0.12 (−1.56–1.32)	0.87	5.55 (0.09–363)	0.42
C18:0	Stearic acid	0.01 (−0.04–0.06)	0.57	1.02 (0.89–1.18)	0.77
C22:0	Behenic acid	−0.09 (−1.45–1.26)	0.89	0.43 (0.01–25.3)	0.69
**Monounsaturated fatty acids**		0.02 (<0.01–0.04) *	0.05	1.11 (1.04–1.18) *	0.002 *
C14:1	Myristoleic acid	0.31 (−0.65–1.27)	0.53	8.09 (0.49–113)	0.14
C15:1	Pentadecanoic acid	0.34 (−2.71–3.39)	0.83	33.1 (0.01–2.2 × 10^5)^	0.44
C16:1n-7	Palmitoleate	0.06 (−0.05–0.17)	0.30	1.39 (1.02–1.89) *	0.04 *
C18:1n-9	Oleic acid	0.02 (<−0.01–0.05) *	0.07	1.10 (1.03–1.19) *	0.009 *
C18:1n-7	Cis-vaccenic acid	0.16 (−0.13–0.45)	0.28	2.87 (1.24–6.64) *	0.01 *
C20:1n-9	Eicosenoic acid	0.06 (−0.55–0.67)	0.85	2.99 (0.56–16.0)	0.20
**Trans fatty acids**		−0.21 (−0.52–0.11)	0.20	0.72 (0.28–1.82)	0.48
C16:1t	Palmitoleic acid	−0.35 (−3.04–2.35)	0.80	34.0 (0.02–7.0 × 10^4^)	0.37
C18:1t	Elaidic acid	−0.23 (−0.56–0.11)	0.19	0.63 (0.23–1.70)	0.36
C22:1t	Erucic acid	−0.02 (−1.92–1.89)	0.99	2.80 (0.01–694)	0.71
**Polyunsaturated fatty acids**		0.01 (−0.01–0.03)	0.50	1.03 (0.96–1.10)	0.45
C20:3n-9 ^5^	Mead acid	−2.04 (−11.09–7.02)	0.66	0.05 (<0.01–1.4 × 10^10^)	0.82
**N-3 polyunsaturated fatty acids**		0.02 (−0.11–0.15)	0.73	1.15 (0.79–1.67)	0.47
C18:3n-3	Alpha-linoleic acid	0.07 (−0.15–0.29)	0.52	1.11 (0.59–2.10)	0.74
C20:3n-3 ^5^	Eicosatrienoic acid	−2.12 (−7.23–2.99)	0.42	20.1 (<0.01–5.9 × 10^7^)	0.69
C20:5n-3 ^5^	Eicosapentaenoic acid	0.04 (−0.84–0.91)	0.94	2.14 (0.18–25.2)	0.55
C22:5n-3 ^5^	Docosapentaenoic acid	−0.16 (−1.24–0.91)	0.77	2.19 (0.10–48.7)	0.62
C22:6n-3 ^5^	Docosahexaenoic acid	<0.01 (−0.29–0.29)	>0.99	1.40 (0.62–3.17)	0.41
**N-6 polyunsaturated fatty acids**		0.01 (−0.02–0.03)	0.50	1.03 (0.96–1.10)	0.49
C18:2n-6	Linoleic acid	0.01 (−0.02–0.03)	0.47	1.02 (0.95–1.10)	0.54
C18:3n-6	Gamma-linoleic acid	0.49 (−1.12–2.10)	0.55	4.57 (0.04–496)	0.53
C20:2n-6	Eicosadienoic acid	−0.11 (−0.53–0.30)	0.60	1.54 (0.47–5.02)	0.48
C20:3n-6 ^5^	Dihomo-gamma-linoleic acid	0.23 (−0.56–1.02)	0.57	5.61 (0.59–53.7)	0.14
C20:4n-6 ^5^	Arachidonic acid	−0.06 (−0.95–0.83)	0.89	1.33 (0.10–17.8)	0.83
C22:4n-6 ^5^	Adrenic acid	−2.17 (−4.51–0.16) *	0.07	1.22 (<0.01–1.5 × 10^3^)	0.96
C22:5n-6 ^5^	Osbond acid	0.45 (−3.70–4.59)	0.83	0.48 (<0.01–1.1 × 10^5^)	0.91
Long chain polyunsaturated fatty acids ^6^		−0.01 (−0.17–0.16)	0.93	1.27 (0.80–2.02)	0.31
Medium chain fatty acids ^7^		−0.02 (−0.04–<0.01) *	0.02 *	0.92 (0.86–0.98) *	0.006 *
Saturated fatty acids/Polyunsaturated fatty acids ^8^		−0.05 (−0.12–0.03)	0.25	0.77 (0.60–0.99) *	0.04 *
n-6/n-3 polyunsaturated fatty acids ^8^		<0.01 (−0.03–0.04)	0.90	1.00 (0.90–1.10)	0.97

^1^ Per one percent increase in the concentration of the respective triglyceride fatty acid. ^2^ Infant growth defined as change in weight-for-age z-score between 0 and 6 months. Models were run in the multiply imputed data set (*n* = 789) in 10 imputed sets and adjusted for maternal age, smoking, education, pre-pregnancy body mass index, gestational weight gain, parity, and child sex. Associations with *p* < 0.05 are denoted with *. ^3^ Rapid growth defined as change in weight-for-age z-score between 0 and 6 months > 0.67. Models were run in the multiply imputed data set (*n* = 789) in 10 imputed sets and adjusted for maternal age, smoking, education, pre-pregnancy body mass index, gestational weight gain, parity, and child sex. Associations with *p* < 0.05 are denoted with *. ^4^ Also classified as a medium chain fatty acid. ^5^ Also classified as a long chain polyunsaturated fatty acid. ^6^ This category includes C20:3n-3, C20:3n-6, C20:3n-9, C20:4n-6, C20:5n-3, C22:4n-6, C22:5n-6, C22:5n-3, and C22:6n-3. ^7^ This category includes C8:0, C10:0, C12:0, and C14:0. As the classification of myristic acid (C14:0) is ambiguous, for the purposes of this analysis, myristic acid is characterized as a medium-chain fatty acid (MCFA) due to its production in the mammary gland along with caprylic acid (C8:0), undecanoic acid (C10:0), and lauric acid (C12:0) and similar effects as other MCFAs in infant diets. ^8^ Exposure used is the ratio, not the percent composition.

## Data Availability

The data presented in this study are available on request from the corresponding author. The data are not publicly available due to privacy and ethical concerns in this ongoing birth cohort.

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
