# Peer review of "Predictors of Human Milk Fatty Acids and Associations with Infant Growth in a Norwegian Birth Cohort"

_nutrients, 2022, doi:10.3390/nu14183858_

Round 1

Reviewer 1 Report

This study was conducted with aims 1) to characterize the composition of fatty acids in human milk, 2) to examine the associations between the maternal factors and fatty acids concentration, and 3) to explore the association of fatty acids in human milk during 14 to 60 days with infant growth in a Norwegian population. Mothers’ factors, such as pregnancy weight gain, parity, gestational age, obesity influenced milk fatty acids composition, and medium chain fatty acids and monounsaturated fatty acids were associated with infant growth and rapid growth negatively and positively, respectively. These results are interesting, but the effect sizes shown in Tables 3, and 4 are not interpretable due to the following major concerns. Commented points may change the results.

Major points

L173 “TGFA concentrations were highly correlated: 12% had Spearman’s r>0.8 (Supplemental Figure 2),” and supplementary table 3 shows all correlations are positive (>0). The authors had a purpose to characterize and use the composition of fatty acids. In composition data, one component concentration increases as some other components concentrations decrease. This relationship must show negative correlations between some components (<0). Supplementary Figure 2 and Table 3 should be depicted based on the percentages of fatty acids. At the same time, the percentages of fatty acids do not have a normal distribution because of their range between 0 and 1. Variables of fatty acids are needed to be log-transformed before linear regression analysis.

Table 2. Concentrations and percentages of fatty acids in milk (n=789). This seems useful for readers. But they want to know the data for general populations. Overweight and obese mothers were oversampled (L73, and L170). Fatty acids concentrations and percentage should be presented as estimated concentrations and percentages using sampling weight.

TGFA composition was measured by gas chromatography (L82). When using gas chromatography, samples may be diluted for appropriate measurement of each target. Concentrations of fatty acids vary from <0.01 g/L for 1st quartile of osbond acid to 9.35 g/L for 3rd quartile of oleic acid; there is a 1000-time difference. For these fatty acids, coefficients of variation ranged from 1% to 33% (L98). Fatty acids with low concentrations may have high coefficients of variation. It is not easy to compare and interpret effect sizes (β coefficients and odds ratios) in Tables 3 and 4 among models for different fatty acids. Considering comments above, variables of log-transformed fatty acids percentages are reduced preliminary using principal component analysis, factor analysis, or latent cluster analysis. This maneuver could put synergistic and antagonistic effects between fatty acids pairs into the models. Inter-fatty-acid effect is another concern.

In Table 4, β coefficients and odds ratios for 1% increase in the concentration of the respective triglyceride fatty acid are shown. For example, rapid growth risk increases 2.87 times as cis-vaccenic acid concentration increases by 1%. However, in Table 3, cis-vaccenic acid has median 1.6%, and IQR 1.5–1.8%; in real human milk, 1% change may seldom appear. Units of independent variables should be one-standard deviations or realistic percentages. Furthermore, ratios for saturated fatty acids/ polyunsaturated fatty acids, and n-6/n-3 polyunsaturated fatty acids may not have percentage units.

In multivariate models in L210–216, and Table 4, adjustment for maternal age, smoking, education, pre-pregnancy body mass index, gestational weight gain, parity, and child sex was done. In nutritional epidemiology, total energy, and other energy-generating nutrients such as protein (L85) are considered. Sometimes substitutional analysis is conducted. The author should consider this point for analysis or discussion.

Minor points

Fatty acids groups (L103–1008) should be explained in detail in addition to footnotes. Explanation is needed in the text or another table.

The authors’ concept was shown in Supplementary Figure 1. However, the paragraph L257265 in section 4.2 is confusing due to inverse causality. The paragraphs in section 4.3 are also confusing because n3 fatty acids did not influence infant growth. The concept in Supplementary Figure 1 suggests structural construction modeling. This is alternative analysis for combination of principal component analysis and regression analysis. Consider it.

Reviewer 2 Report

In this study authors aimed to characterized the concentrations and percent composition of detectable TGFAs in human milk and studied the relation between maternal factors and these TGFAs. Additionally they explored associations between TGFA milk composition and infant growth during the first six months of life. This is a valuable contribution to the field, however the authors may be encorraged to improve the work by adding some minor aspects:

Line 173 “TGFA concentrations were highly correlated” with between??? Please rephrased it.

Line 242: Medium-chain saturated fatty acids were inversely associated with growth::: infant growth?? Please rephrased it.

Lines 234-255  please provide more holistic data of MCFAs, from structural characteristics to metabolic effects in infants.

Round 2

Reviewer 1 Report

Thank you for your revision. Several figures improve the manuscript.

In the Discussion (Lines 276–277, and 326–3338), and the Abstract (Lines 27–29), cis-vaccenic acid (C18:1n-7) was focused on. However, within the conceivable range of each fatty acid (the interquartile range), rather than in 1% change (Table 5), oleic acid (C18:1n-9) has a larger effect size than cis-vaccenic acid and palmitoleic acid. It is better to focus on and discuss oleic acid more in detail.

Table 5. Asterisks (*) indicates p<0.05, according to the footnotes. But for infant growth, p values of oleic acid, and adrenic acid are 0.07 with asterisks, p values of SFAs, and MUFA are 0.05, which is not less than 0.05. The confidence intervals of these estimates do not show significant range.

Figure 3. there are three blue dots (significant), but there are only two labels. The last one may be C12:0.??

In the legend, “Each dot represents a beta coefficient per interquartile range (IQR) increase in TGFA composition. X-axis represents change in weight-for-age z-score. “ a little bit confusing. For example, “each dot represents change in weight-for-age z-score (x axis) per interquartile range (IQR) increase in TGFA.”???

Figure 4. Does the label “MCFA, c12_0” mean that two dots completely overlap (or it explains that c12_0 is MCFA)? It may be explained in the legend, or using each line with each label on the plot. In the same way as above, it is not easy to understand that “beta coefficient” and “log of odds ratio (OR) for rapid growth” are the same.

Several fatty acids are written in two styles, -ic acid, and -ate. These should be consistent in an article.
